# Effect of Temperature and Impurity Content to Control Corrosion of 316 Stainless Steel in Molten KCl-MgCl_2_ Salt

**DOI:** 10.3390/ma16052025

**Published:** 2023-02-28

**Authors:** Na Li, Huaiyou Wang, Huiqin Yin, Qi Liu, Zhongfeng Tang

**Affiliations:** 1Shanghai Institute of Applied Physics, Chinese Academy of Sciences, Shanghai 201800, China; 2University of Chinese Academy of Sciences, Beijing 100049, China; 3Qinghai Institute of Salt Lakes, Chinese Academy of Sciences, Xining 810008, China; 4Dalian National Laboratory for Clean Energy, Dalian 116023, China

**Keywords:** corrosion, high temperature, KCl-MgCl_2_, molten salt, alloy

## Abstract

The corrosion resistance of 316 stainless steel (316SS) in molten KCl-MgCl_2_ salts was studied through static immersion corrosion at high temperatures. Below 600 °C, the corrosion rate of 316SS increased slowly with increasing temperature. When the salt temperature rises to 700 °C, the corrosion rate of 316SS increases dramatically. The corrosion of 316SS is mainly due to the selective dissolution of Cr and Fe at high temperatures. The impurities in molten KCl-MgCl_2_ salts could accelerate the dissolution of Cr and Fe atoms in the grain boundary of 316SS, and purification treatment can reduce the corrosivity of KCl-MgCl_2_ salts. Under the experimental conditions, the diffusion rate of Cr/Fe in 316SS changed more with temperature than the reaction rate of salt impurities with Cr/Fe.

## 1. Introduction

KCl-MgCl_2_ eutectic salt has excellent thermophysical properties [1,2]. Its melting point is approximately 426 °C, and its thermal conductivity is about 0.4 W·m^−1^·k^−1^ [3,4]. The vapor pressure of molten KCl-MgCl_2_ salt is lower than 13 kPa even at 800 °C [4]. In addition to its excellent heat transfer performance, the price of KCl-MgCl_2_ salts is not high [5], so it is considered one of the best heat transfer and storage mediums in next-generation concentrated solar power (CSP) and high-temperature reactors [6,7,8]. KCl-MgCl_2_ molten salts should also have good compatibility with heat storage containers, pipelines, and valves to realize their application as heat transfer and storage medium [3,9]. In order to explore the interaction between molten KCl-MgCl_2_ salts and alloys, the corrosion behavior and mechanism have been investigated in the literature [10,11,12]. Similar to the NaCl-KCl-MgCl_2_ [13,14,15], NaCl-MgCl_2_ [16,17,18], NaCl-KCl-CaCl_2_ [19], NaCl-KCl-ZnCl_2_ [20], and nitrate salts [21,22], the corrosion control of the alloys in molten KCl-MgCl_2_ salts is a challenging task.

The impurities in molten salts were the main reason for the alloy corrosion [23]. Purification of molten chlorides helped to remove dissolved oxide and reduced the corrosion of salt [6]. Purification of molten chlorides usually uses one of three methods: high-temperature drying, chemical purification, and metal reduction. The corrosion performances of Hastelloy C-276 and Haynes 230 corroded in 800 °C molten KCl-MgCl_2_ salts were weak after pretreatment at 500 °C [1]. The corrosion rate of 316 L and N alloys in 700 °C molten KCl-MgCl_2_ salts was reduced using carbon tetrachloride as the chlorination reagent to reduce dissolved oxide [24]. The corrosivity of molten KCl-MgCl_2_-based salts is reduced because Mg treatment removes corrosive impurities [25,26]. Similar to other molten MgCl_2_-based salts, the oxygen in the molten salts will react with MgCl_2_ to form MgO at high temperatures [27]. The dense MgO film formed on the alloy surface can slow down the alloy corrosion, and the molten salt temperature has a great influence on the stability of the MgO protective layer. High temperature damages the MgO protective layer, which will accelerate the corrosion [11]. For CSP equipment, a high working temperature can realize the more efficient conversion of thermal energy to electrical energy and faster corrosion [4,28]. However, the higher the temperature is, the faster the corrosion of the alloy [27]. Under the same conditions, the corrosion rate of 316SS at 500 °C is less than 1/20 of that at 700 °C [17]. Therefore, it is very important to select the appropriate operating temperature of molten KCl-MgCl_2_ salts for practical applications. 

The working temperature (500–720 °C) of molten salts was proposed by NREL in the next-generation CSP [6]. However, the effect of salts temperature on the alloy’s corrosion in molten KCl-MgCl_2_ salts is mainly concentrated above 700 °C, and there is no research on its corrosion performance at 500–700 °C. In addition, the coupling effect of temperature and impurity ion concentration in molten salt on the alloy’s corrosion has not been studied. In this work, the influence of the salt purification and temperature on the corrosion of 316 stainless steel (316SS) was investigated through static immersion in molten KCl-MgCl_2_ eutectic salts at 500 °C–700 °C. These results are helpful in accurately revealing and controlling the corrosion of 316SS in molten KCl-MgCl_2_ salts.

## 2. Experimental

### 2.1. Materials

KCl and MgCl_2_ (AR, 99.8 wt%) were purchased from Sinopharm Chemical Reagent Group Co., Ltd. (Shanghai, China). KCl-MgCl_2_ eutectic salts (A, 62.5–37.5 wt%) were prepared by physical mixing and melting [16,17]. Purified KCl-MgCl_2_ eutectic salts (B, 62.5–37.5 wt%) were prepared by adding Mg reducing agent and dehydrating at 600 °C for 24 h from Shanghai Institute of Applied Physics, Chinese Academy of Sciences. The whole purification process was carried out in high-purity Ar [16,17]. The cationic impurities of molten KCl-MgCl_2_ eutectic salts were determined by inductively coupled plasma-optical emission spectroscopy (ICP-OES, Spectro, Kleve, Germany). The anion impurities in molten KCl-MgCl_2_ salts were analyzed by ion chromatography. The residual H_2_O and oxygen in molten KCl-MgCl_2_ salts were determined by Karl-Fischer titration and LECO oxygen analyzer. The concentrations of residual H_2_O and oxygen in salt A were 300 ppm and 1520 ppm, and the concentrations of the corresponding substances in salt B were 10 ppm and 90 ppm, respectively. The main impurities concentration in salts A and B are shown in Table 1.

The 316SS alloys were prepared as described in reference [17]. The size of 316SS sheets is 15.0 mm × 10.0 mm × 3.0 mm. The main components of 316SS are listed in Table 2. The marked 316SS specimens were cleaned, dried, and weighed in this experiment. 

### 2.2. Static Corrosion Test

In Ar (99.995%), the static corrosion experiment of 316SS in salt A or B was conducted on a furnace connected to a glove box. The schematic diagram of the experimental device is shown in Figure 1. The Al_2_O_3_ crucibles were used for the corrosion test. These crucibles were cleaned and dried at 700 °C for 8 h before corrosion tests. Three specimens of 316SS were put in an Al_2_O_3_ crucible containing 75.0 g salt. Static corrosion of 316SS alloys was measured at 500 °C, 600 °C, and 700 °C for 100 h in Ar, respectively. All operations were performed in the glove box filled with Ar, where the water and oxygen were controlled below 10 ppm by a gas purification system. To clean the corroded 316SS alloys, they were treated with deionized water, rinsed with anhydrous alcohol, and dried with cold air.

### 2.3. Characterisation

Weight changes caused by the corrosion of 316SS can be obtained by the following formula [17].
(1)$$\Delta m=\frac{m_{a}-m_{b}}{S}$$
where Δm is the mass change per unit area (mg/cm^2^), $m_{a}$ and $m_{b}$ stand for the mass of 316SS pre- and post-corrosion, respectively. S is the surface area of alloy pre-corrosion. Weight change is the mean of three parallel 316SS alloys. 

Scanning electron microscopy (SEM, Merlin Compact, ZEISS, Oberkochen, Jena, Germany) coupled with an energy-dispersive X-ray spectrometer (EDS) was used to analyze the surface and cross-sectional morphology changes of the corroded 316SS. After cleaning and drying, the surface morphology and element analysis of the alloy specimens could be carried out directly. Before the cross-section analysis, alloy specimens need to be embedded with cold-curing epoxy resin, then ground to 1200 grit with silicon carbide paper and polished with 0.05 μm Al_2_O_3_ powder. After cleaning and drying, the cross-section morphology and element analysis of alloy samples could be carried out. The crystal phases of the cleaned and dried 316SS specimens before and after corrosion were measured by X-ray diffraction (Bruker D8 Advance, Bruker, Karlsruhe, Germany). The impurities of Cr and Fe in exposed salt were analyzed by ICP-OES (Spectro ARCOS, Spectro, Kleve, Germany).

## 3. Results and Discussion

### 3.1. Weight Change 

Weight changes of 316SS pre- and post-corrosion in salt A or B at 500, 600, and 700 °C for 100 h under Ar were shown in Figure 2. The corrosion weight loss of 316SS was positively related to the temperature increase. However, the weight loss of the 316SS was not linearly related to the temperature increase. Below 600 °C, the corrosion weight loss of 316SS in salt A was less than 1.28 mg/cm^2^ and increased slowly with increasing temperature. When the temperature of salt A increased from 500 °C to 600 °C, the corrosion weight loss increased by only 0.73 mg/cm^2^. At the same time, when the salt temperature increased from 600 °C to 700 °C, the corrosion weight loss of 316SS increased sharply by 1.84 mg/cm^2^.

It was also confirmed that the purification of molten chloride salts was an effective method of reducing corrosion [18]. The weight loss of corroded 316SS in salt B was lower than that of salt A at the same temperature. Although the purification of molten KCl-MgCl_2_ salt affected the corrosion of 316SS, the rule of corrosion weight loss with salt temperature did not change due to the salt purification.

### 3.2. Microscopic Morphology Analysis

Surface images of 316SS pre- and post-corrosion in salts A and B were characterized by SEM, as shown in Figure 3. After immersing in salt A (A_1_, A_2_, A_3_) at 500, 600, and 700 °C for 100 h, grain boundary corrosion occurred on the surface of corroded 316SS. With the increase in salt temperature, the grain boundary of 316SS alloy surface corrosion became more and more obvious. Although the surface corrosion of 316SS in salt B was weaker than that in salt A, the grain boundary corrosion tendency of 316SS did not change with the temperature increase. At 500 °C or 600 °C, grain boundary corrosion marks on the surface of 316SS surface were relatively shallow. When the temperature of molten KCl-MgCl_2_ salts reaches 700 °C, whether molten KCl-MgCl_2_ salts were purified or not, the grain boundary corrosion on the surface of 316SS was obvious after being immersed in molten KCl-MgCl_2_ salts for 100 h. When the temperature exceeded 600 °C, the grain boundary corrosion on the surface of 316SS deteriorated sharply as the temperature rose to 700 °C.

Cross-sectional images of 316SS before and after being corroded in salts A and B for 100 h at three different temperatures were shown in Figure 4. 316SS alloys were slightly corroded, and the surface began to appear uneven in 500 °C salts B. However, after corrosion in non-purified salt, some grain boundary corrosion could be seen in the surface layer of the cross-section. When the temperature rose to 600 °C, grain boundary cracks appeared in the cross-section of corroded 316SS, regardless of whether it was purified or not. When the corrosion temperature rose to 700 °C, the intergranular corrosion became more obvious, and the corrosion speed was accelerated. There were many large cavities on the cross-section, and the cracks at the grain boundaries widened and extended into the matrix. 

The distribution depth of corrosion holes in the 316SS cross-section was also quantitatively described in Figure 4. The relationship between corrosion hole depth and temperature in salt A is similar to that in salt B, except that the distribution depth of corrosion holes of the 316SS is deeper than that in salt B.

### 3.3. Element Distribution Analysis

Surface morphology and composition of 316 SS (B_2_) after being corroded in purified KCl-MgCl_2_ molten salt at 500 °C (B1), 600 °C (B2), and 700 °C (B3) under Ar were shown in Figure 5. After being exposed to molten salts for 100 h, obvious grain boundary corrosion can be observed. Fe and Cr are depleted at grain boundaries, while O and Mg are enriched.

In order to analyze the influence of salt purity and temperature on the corrosion behavior of 316SS, the Cr depletion depth of 316SS after being exposed to unpurified and purified KCl-MgCl_2_ salt under Ar for 100 h was also measured by SEM-EDS in line scanning mode, and the results were shown in Figure 6.

The Cr depletion depths of the corroded 316SS exposed in salt A are 5.2 µm, 12.1 µm, and 30.4 µm at 500 °C, 600 °C, and 700 °C, respectively. The effect of temperature on Cr depletion depth increased with the increase in temperature. The Cr depletion depth of 316SS corroded in purified molten salts also increased with the increase in temperature. The depletion depth of salt B increased from 3.3 µm at 500 °C to 8.7 µm at 600 °C. When the salt temperature reached 700 °C, the depletion depth reached 23.6 µm. However, compared with the corrosion performance of unpurified salt, it can be found that the corrosion ability of the purified salt decreased, and the Cr depletion depth decreased at the same temperature. 

The rule of Fe depletion depths is also similar to that of corrosion hole depth. With the increase in temperature, the Fe depletion depth of the corroded 316SS increases, and salt purification can reduce the Fe depletion depth of the 316SS surface.

### 3.4. Crystalline Structure Variation 

XRD patterns of the original and corroded 316SS in the unpurified and purified KCl-MgCl_2_ molten salts at different temperatures are shown in Figure 7. Before corrosion, the characteristic diffraction peaks of 316SS, as shown in the black XRD pattern, indicate that the initial microstructure of 316SS alloy is a single-phase face-centered cubic (FCC) austenite structure. After corrosion, the crystal structure of the corroded 316SS remained austenite structure. Only the diffraction peaks of the 316SS corroded in salt slightly moved to a higher angle, which is attributed to the dissolution of Fe and Cr on the surface of 316SS [14,27]. Except for the matrix peak of 316SS, a new characteristic peak appeared on the surface of the alloy after corrosion. Combined with the analysis of SEM, it could be inferred that it was the characteristic peak of MgO. This shows that MgO appeared on the surface of the corroded alloy.

The results of XRD show that the basic crystal structure of the 316SS alloy surface will not change with the change of temperature and purity after corrosion. However, when the salt temperature and salt purity change, the shift of the characteristic peak position of the 316SS alloy matrix to a high angle is slightly different.

### 3.5. Analysis of Corrosion Products

The content and concentration of impurities in molten salts greatly influence their corrosion performance. The impurities in the molten salts, such as water, oxygen, and other oxides, will dissolve Cr and Fe atoms on the surface of 316SS and enter the molten salts, thus accelerating the corrosion of the surface of 316SS [28,29]. Cr and Fe ion content in the unpurified and purified KCl-MgCl_2_ molten salts at pre- and post-corrosion are shown in Figure 8. Cr and Fe ion content in the unpurified and purified salts was very low pre-corrosion. The concentration of Cr and Fe ions in the KCl-MgCl_2_ molten salts increases with the corrosion temperature increases, which indicates that the corrosion rate of the 316SS accelerated with the increase in temperature. Compared with the concentration of Cr and Fe ions in molten KCl-MgCl_2_ salts before the corrosion experiment, the Cr ion concentration in the unpurified and purified KCl-MgCl_2_ salt post-corrosion increased by 0.11 mg/kg and 0.08 mg/kg at 500 °C after 100 h, respectively. The Cr ion concentration in the unpurified and purified KCl-MgCl_2_ salt post-corrosion increased by 0.15 mg/kg and 0.11 mg/kg at 600 °C after 100 h, respectively. When molten KCl-MgCl_2_ salt temperature is lower than 600 °C, the increase deviation of the two is less than 0.05 at the same temperature. When the molten KCl-MgCl_2_ salts temperature rises to 700 °C, the Cr ion concentration in salt A reaches 0.38 mg/kg, while the Cr ions concentration in salt B is only 0.25 mg/kg, and the deviation has reached 0.13 mg/kg. The increasing rule of Fe ion is the same as that of Cr ion. The increasing rule of Fe ion concentration in molten KCl-MgCl_2_ salts at post-corrosion is the same as that of Cr ion under the same conditions. The concentration of Cr and Fe ions in salt A is higher than in salt B. With the temperature increase, the increase in Cr and Fe ions in salt B is quite different from that in salt A, and the purification effect of molten KCl-MgCl_2_ salts is more obvious. The Arrhenius formula is used for data analysis, as shown in Figure 8, and the apparent corrosion activation energy of Cr and Fe in salts B were 5.5 kJ/mol and 5.6 kJ/mol, respectively. The activation energy of Fe was slightly higher than that of Cr.

### 3.6. Corrosion Mechanism

Impurities in molten KCl-MgCl_2_ salts could lead to the corrosion of 316SS [23]. Due to it containing trace amounts of water, oxygen, and other impurities, MgCl_2_ salt easily absorbs moisture to form hydroxyl magnesium chloride, which will decompose at high temperatures to form HCl and MgO, as shown in Equations (2) and (3) [15,17]. As shown in Equations (4) and (5), Cr and Fe in the 316SS will react with HCl to generate chloride. Mg treatment in the purification process may remove the corrosive HCl impurities and MgOHCl, which obviously decreases the corrosivity of the molten salts [15].
Mg^2+^ + H_2_O + Cl^−^ → MgOH^+^ + HCl (g)(2)
MgOH^+^ → MgO + HCl (g) (3)
Cr + 2H^+^ → Cr^2+^ + H_2_ (g) (4)
Fe + 2H^+^ → Fe^2+^ + H_2_ (g) (5)

In a chemical reaction, the reaction rate is directly related to the concentration of reactants, and the higher the concentration of oxidizing impurities in salt, the faster the dissolution rate of Cr and Fe. Mg reducing agent was used to purify KCl-MgCl_2_ eutectic salt, which reduced the concentration of oxidation impurities, thus reducing the corrosion of 316 SS. When Cr and Fe in the outermost layer of the 316SS alloy are dissolved, the Cr and Fe concentrations on the surface of the 316SS are different from those in the matrix. Driven by the concentration gradient, Cr and Fe in the matrix diffuse from the matrix to the surface of 316SS and continued to react with impurities in KCl-MgCl_2_ eutectic salt, resulting in continuous corrosion of the alloy. 

A large number of studies showed that the corrosion rate of Cr and Fe is not only related to the concentration gradient but also related to the diffusion rate of Cr and Fe atoms [6,29]. The results show that the grain boundary diffusion rate of Cr in 316SS is about 106 times the lattice diffusion rate, which indicates that grain boundary diffusion is much faster than lattice diffusion [30]. Therefore, after 100 h of corrosion test, all the corroded 316SS samples showed different degrees of grain boundary corrosion.

The diffusion of Cr and Fe in 316SS can be described by [31]:D = D_0_e^(−Q/RT)^
(6)
where D is the diffusion coefficient, D_0_ is the material coefficient, Q is the diffusion activation energy, R is the inert gas constant, and T is the temperature in Kelvin. With the increase in temperature, the diffusion coefficient of atoms increases, and the diffusion rate of Fe and Cr atoms from 316 SS matrix to surface increases, which leads to resulting in the increase in the intergranular corrosion rate of 316SS, which is consistent with the corrosion trend of 316SS in NaCl-MgCl_2_ salt [17].

It is found that impurity content in salt and salt temperature will affect alloy corrosion [17,23,24,25]. The influence of impurities on the corrosion rate of the 316SS alloy was not as significant as that of temperature, which was due to the impurity content in the unpurified KCl-MgCl_2_ eutectic salt was not much higher than that in the purified salt, and the corrosion tests were carried out in an inert atmosphere. The difference in impurities concentration between salts was limited. This might also be the main reason why the effect of purification on 316SS corrosion is less than that of temperature. Meanwhile, the corrosion process of alloy in purified salt is largely determined by the diffusion rate of corrosive atoms such as chromium in the matrix to the outermost layer of alloy, which increases linearly with the increase in temperature [30], so the temperature has a great influence on the corrosion of 316SS.

## 4. Conclusions

The effect of temperature and purity on the corrosion of 316SS in molten KCl-MgCl_2_ salts was studied through static immersion corrosion under Ar at 500 °C, 600 °C, and 700 °C for about 100 h. The results showed that the corrosion rate of 316SS in KCl-MgCl_2_ salt increased nonlinearly with the increase in temperature. Below 600 °C, the corrosion rate of 316SS increased slowly with the increase in temperature. When the salt temperature rose to 700 °C, the corrosion rate of 316SS increased dramatically. Purification can reduce the corrosivity of molten KCl-MgCl_2_ salts. At these three temperature points, the corrosion of 316SS by KCl-MgCl_2_ salt after Mg metal reduction and purification weakened.

Increasing the salt temperature speeds up the diffusion of elements in the alloy while increasing the impurity concentration in the salt accelerates the reaction rate of 316SS elements, both of which will accelerate the corrosion of 316SS. Under the experimental conditions, the diffusion rate of Cr/Fe in the 316SS changed more with temperature than the reaction rate of salt impurities with Cr/Fe, which may be the reason why the effect of purification on the corrosion rate of 316SS in molten KCl-MgCl_2_ salts is less than that of temperature change.

## Figures and Tables

**Figure 1 materials-16-02025-f001:**
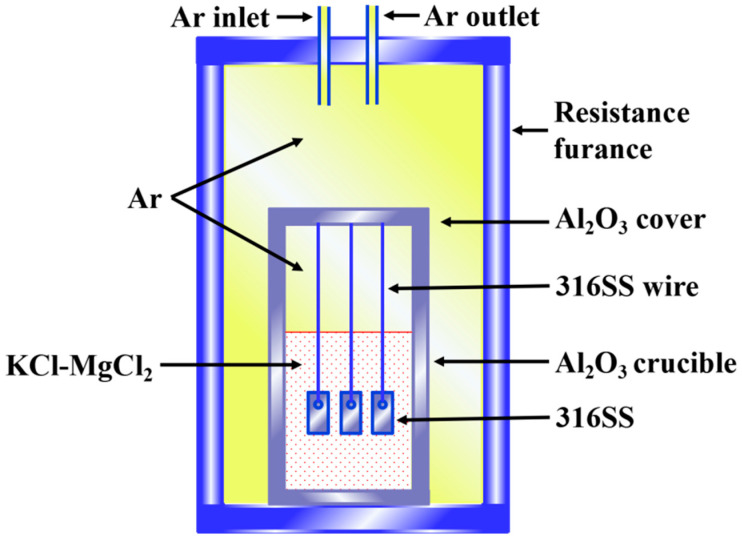
Diagram of the corrosion experimental setup.

**Figure 2 materials-16-02025-f002:**
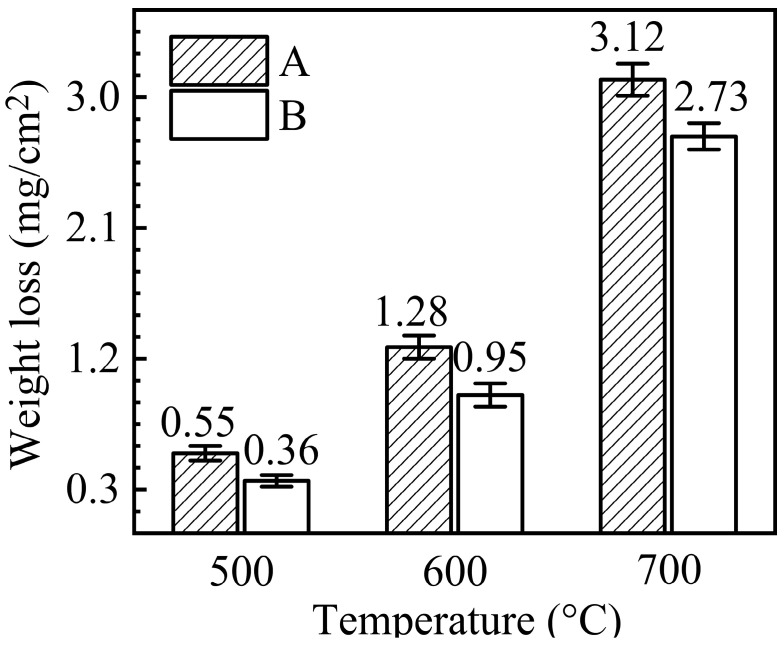
Weight changes of 316SS alloys immersed in salts A and B at 500, 600, and 700 °C under Ar for 100 h.

**Figure 3 materials-16-02025-f003:**
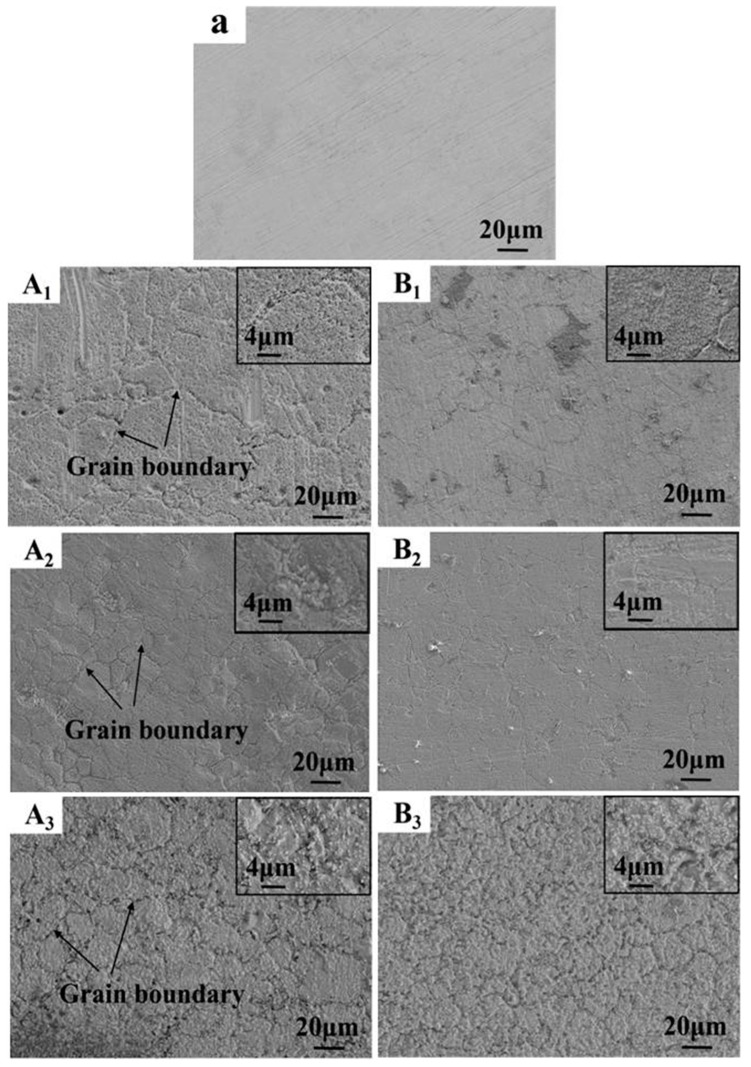
Surface SEM images of original 316SS (**a**), corroded 316SS exposed in salts A (**A_1_**–**A_3_**) and B (**B_1_**–**B_3_**) under Ar for 100 h at 500 °C (**A_1_**, **B_1_**), 600 °C (**A_2_**, **B_2_**) and 700 °C (**A_3_**, **B_3_**).

**Figure 4 materials-16-02025-f004:**
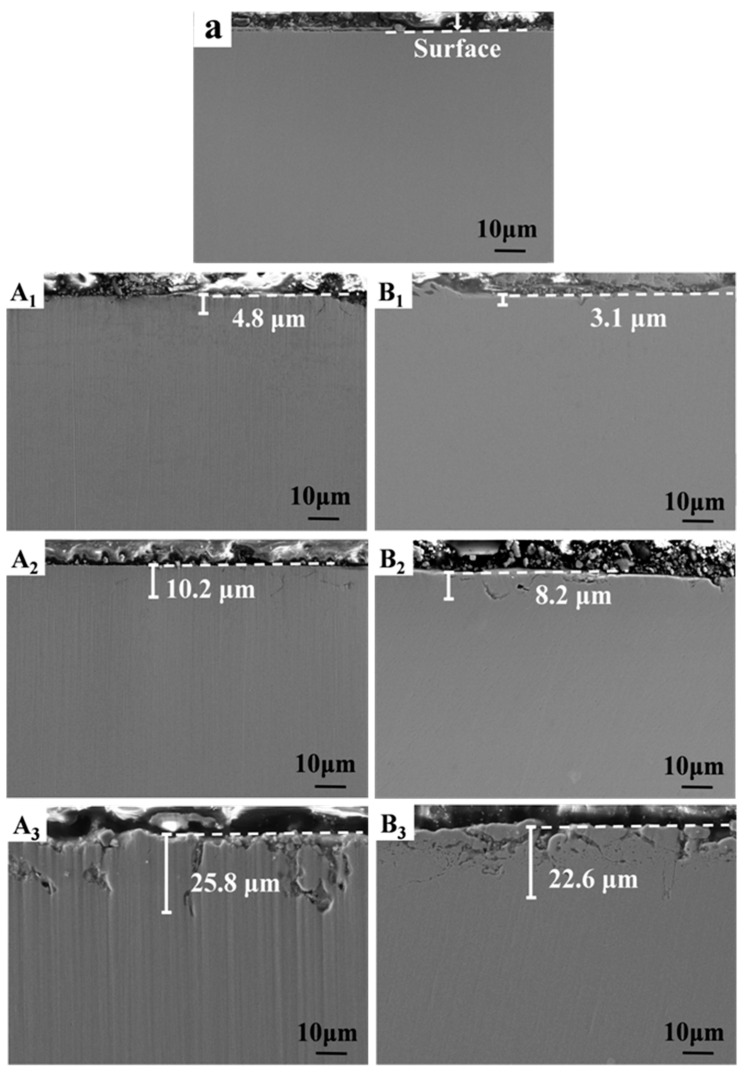
Cross-sectional SEM images of the original 316SS (a), corroded 316SS exposed in salts A (A_1_–A_3_) and B (B_1_–B_3_) under Ar for 100 h at 500 °C (A_1_, B_1_), 600 °C (A_2_, B_2_) and 700 °C (A_3_, B_3_).

**Figure 5 materials-16-02025-f005:**
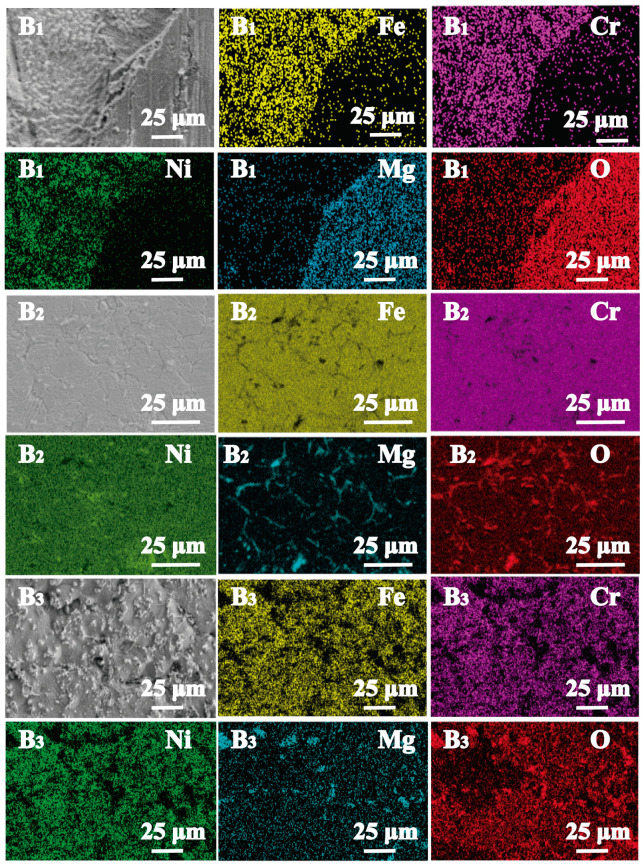
Element distribution of 316SS exposed in purified salt under Ar for 100 h at 500 °C (**B_1_**), 600 °C (**B_2_**), and 700 °C (**B_3_**).

**Figure 6 materials-16-02025-f006:**
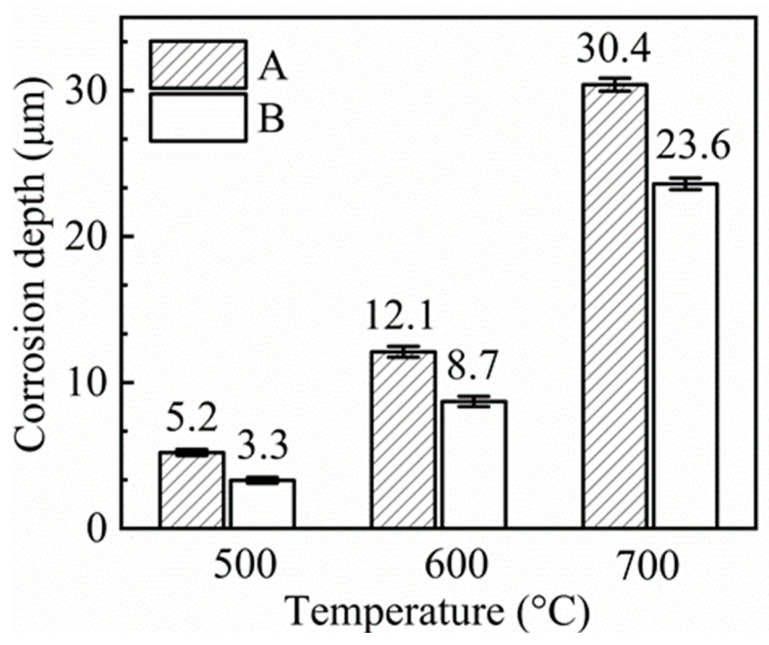
Cr depletion depths of corroded 316SS exposed in salts A and B under Ar for 100 h at 500 °C, 600 °C, and 700 °C.

**Figure 7 materials-16-02025-f007:**
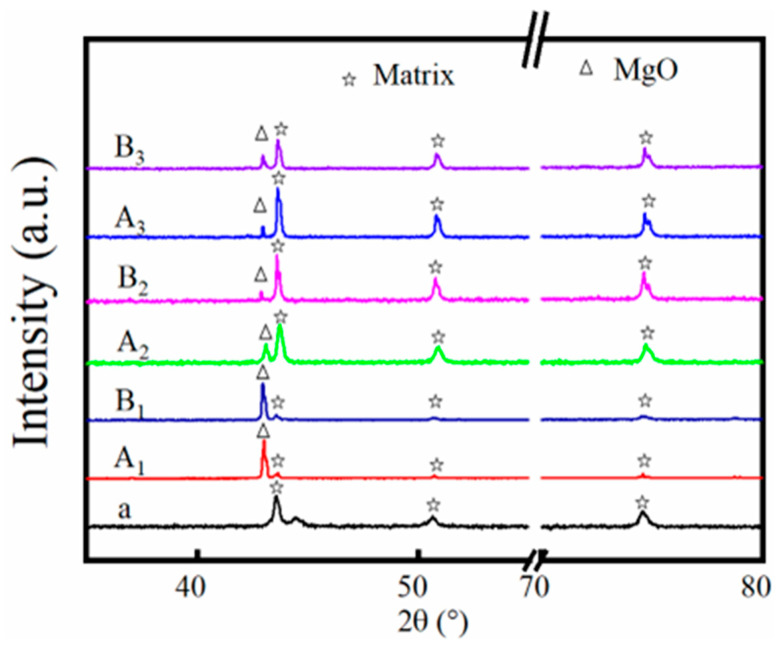
XRD patterns of the original 316SS (a) and corroded 316SS in salts A (A_1_–A_3_) and B (B_1_–B_3_) under Ar for 100 h at 500 °C (A_1_, B_1_), 600 °C (A_2_, B_2_) and 700 °C (A_3_, B_3_).

**Figure 8 materials-16-02025-f008:**
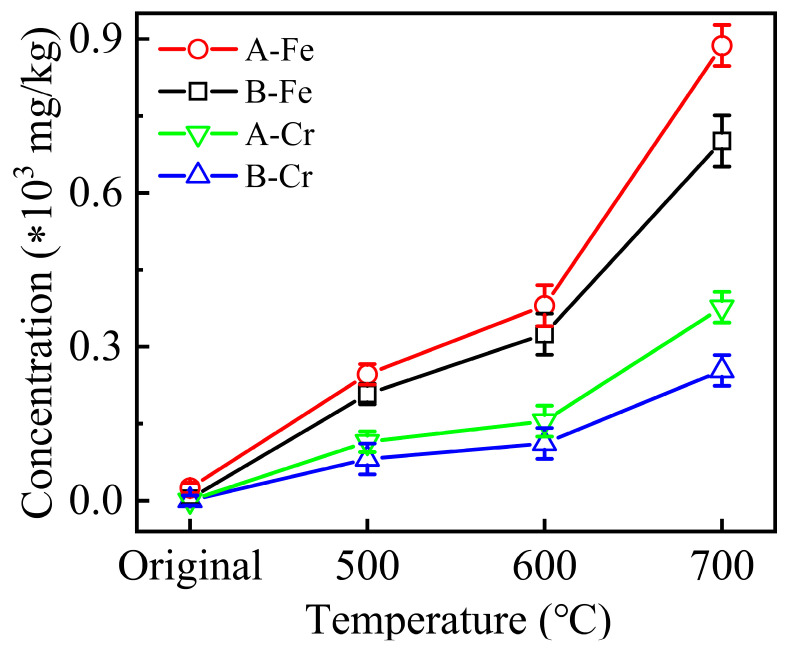
Concentrations of dissolved Cr and Fe elements in the post-corrosion salts A and B at different temperatures for 100 h.

**Table 1 materials-16-02025-t001:** The main impurities in salts A and B pre-corrosion tests (mg/kg).

	Impurity	Fe	Cr	Ni	Mn	Si	SO_4_^2−^	NO_3_^−^	NO_2_^−^	PO_4_^3−^
Salts										
A		24.69	0.38	0.32	0.36	1.35	155.3	188.1	152.4	130.6
B		0.12	N.D	N.D	N.D	0.02	N.D	N.D	N.D	N.D

N.D” is in concentrations below the quantitative limits of detection for ICP-OES.

**Table 2 materials-16-02025-t002:** Chemical compositions of 316SS (wt%).

Alloy	Fe	Cr	Ni	Mn	Si	P	N	Cu	Mo	C
316SS	69.04	16.84	10.12	1.284	0.472	0.021	0.008	0.09	2.08	0.0375

## Data Availability

The data presented in this study are available on request from the corresponding author.

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
