# Peer review of "Effect of Temperature and Impurity Content to Control Corrosion of 316 Stainless Steel in Molten KCl-MgCl2 Salt"

_materials, 2023, doi:10.3390/ma16052025_

Round 1

Reviewer 1 Report

The paper presents the results obtained from corrosion experiments of stainless steel in static immersion at 500°C, 600°C, and 700°C in KCl-MgCl2. The authors entitled the work “Effect of temperature and impurity content to control corrosion of 316 stainless steel in molten KCl-MgCl2 molten salts”; however, is not explained the effect of the temperature or impurities in the corrosion process. Also, some other issues should be improved since the manuscript in its present form is very weak. The details are given below:

1)    The introduction is too summarized; the authors must include more details about the results of the works cited. Please consider that the introduction presents state-of-the-art and the motivation of the research; the results are not included in this section (the last two paragraphs should be removed). In regards to motivation, please clarify why the temperatures of the study are 500, 600, and 700 °C.

2)    In the experimental section, include information about the determination of the chemical composition of the steel.

3)    In the results and discussion section, the analysis is too poor. The discussion must be improved and clarified the effect of the temperature and impurities. In Fig. 2, the results represent the absolute weight loss?

Please include more comparisons with similar works of intergranular and breakaway corrosion mechanisms.

Why are there no mapping images of 500 and 700 °C?

Please include the physical state of the compounds, particularly CrCl2 and FeCl2.

4)    Reformulate the conclusion to clarify the effects of the temperature and impurities of molten salts.

Author Response

  1. The introduction is too summarized; the authors must include more details about the results of the works cited. Please consider that the introduction presents state-of-the-art and the motivation of the research; the results are not included in this section (the last two paragraphs should be removed). In regards to motivation, please clarify why the temperatures of the study are 500, 600, and 700 °C.

Response: Thanks for the reviewer’s comment. Based the reviewer’s suggestion, the introduction was modified. So far, there has not been much research on the corrosion behavior of KCl-MgCl2 salt, and most of these researchers only pay attention to the influence of salt purification on the corrosion behavior of alloys, but do not consider the influence of temperature, which is the key in the application of KCl-MgCl2 salt. In this work, we not only compared the corrosion behavior of salt with different impurity concentration, but also investigated the effect of temperature, and compared the effects of molten salt purification and temperature change on the corrosion behavior of 316SS. The results showed that the effect of temperature change on 316SS corrosion was greater than that of magnesium purification.

  1. In the experimental section, include information about the determination of the chemical composition of the steel.

Response: Thanks for the reviewer’s comment. We use commercial 316SS as the testing material, and its chemical composition comes from the testing report provided by the supplier.

  1. In the results and discussion section, the analysis is too poor. The discussion must be improved and clarified the effect of the temperature and impurities. In Fig. 2, the results represent the absolute weight loss?

Response: Thanks for the reviewer’s comment. We have further modified the test results and presented them in blue font in the revised manuscript. Figure 2 was the mass change per unit area of corroded of 316, which is the average of three parallel samples.

  1. Please include more comparisons with similar works of intergranular and breakaway corrosion mechanisms.

Response: Thanks for the reviewer’s comment. The corrosion mechanism in KCl-MgCl2 is basically the same as that of most chloride salts, the 316SS samples also show the dissolution of Fe and Cr at grain boundaries. The discussion part has been explained and supplemented accordingly.

  1. Why are there no mapping images of 500 and 700 °C?

Response: Thanks for the reviewer’s comment. Since the morphologies of all corrosion samples have been shown in Figures 3 and 4, Figure 5 does not continue to show these results, but only selects the element distribution on the surface of corrosion sample at 600℃ as a representative to illustrate the grain boundary depletion of iron and chromium and the enrichment of magnesium oxide on the surface of corrosion sample.

  1. Please include the physical state of the compounds, particularly CrCl2and FeCl2.

 Response: Thanks for the reviewer’s comment. ICP can analyze the total amount of iron ions and chromium ions, but can't judge the valence state of ions. According to our previous research, iron and chromium mainly exist in the form of bivalent in the corroded KCl-MgCl2 salt.

  1.  Reformulate the conclusion to clarify the effects of the temperature and impurities of molten salts.

 Response: Thanks for the reviewer’s comment. Based the reviewer’s suggestion, the conclusion was modified and presented in blue font.

Reviewer 2 Report

I can recommend the publication of this manuscript after a minor revision.

1. Lines 11-12, write the correct e-mail for every co-author.

2. Write keywords in alphabetical order.

3. The writing must be improved, it is still very poor with numerous wrong terms, typos, or grammar mistakes (i.e. line 73, line 151, and so on).

4. Lines 28-36: .....[4-8]........  [9-16].....[17-21]..... [23-27]......

You will likely need to re-write your citation sentences, rather than simply replace the numbers with Authors’ names. This is due to the fact that in order to give readers the maximum appreciation of how your work builds on previous results, each one of the cited sources should be discussed individually and explicitly to demonstrate their significance to your study. We ask that you use the authors' surnames as the subject of a verb, and then state in one or two sentences what they claim, what evidence they provide to support their claim, and how you evaluate their work. We also, therefore, ask that you avoid citing more than one reference in one sentence. This will give you a chance to discuss each reference separately.

What we are asking for is something like this: “Smith (2011) describes the development of a finite element model of hot forging and claims excellent agreement between the model and experiments.  However, he tests only one operating condition, tunes his model by modifying the friction coefficient, and compares only the total tool force. A much more detailed comparison would be required to evaluate the precise conditions under which finite element modeling is truly accurate."

5. Insert a picture with the “Static corrosion equipment” (line 88).

6. Line 94. Erase “equation-1” and write after the mathematical formula only equation number (1).

7. Line 95: erase (unit:..). It is not necessary.

8. Lines 98-103. Write complete information about tools (country, and so on).

9. Give more details about the statistical analysis applied in this manuscript (method, software, validation, and so on).

10, Lines 125-126: What does it mean “relatively smooth”. Can you specify some roughness parameters for this surface microtexture?

11. Insert all the SEM parameters, such as magnification, acceleration voltage, working distance, and image pixel resolution (figs. 3 and 4).

12. Explain with more details sentences from pages: 148-149, 174-175, 252-255.

13. Lines 231-234. Erase “equation-2....5” and write after the mathematical formula only the equation number in parentheses.

14. Insert Author Contributions, Data Availability Statement, and Conflict of interest.

15. Even though the work is relevant to the journal's scope, i.e., Materials, I do not find even a single article published in the journal in the list of references. Please, some references from MDPI journals.

Authors may consider citing the following references:

[1] DOI: 10.1016/j.corsci.2022.110561

[2] https://doi.org/10.1021/acsami.9b19099

[3] Åžtefan Ţălu, Micro and nanoscale characterization of three-dimensional surfaces. Basics and applications. Napoca Star Publishing House, Cluj-Napoca, Romania, 2015.

This manuscript can be published after the mentioned revisions.

Author Response

  1. Lines 11-12, write the correct e-mail for every co-author.

Response:Thank you for your suggestion. According to your suggestion, we checked the email address of corresponding authors in the revised manuscript.

  1. Write keywords in alphabetical order.

Response:Thank you for your suggestion. According to your suggestion, we write keywords in alphabetical order in the revised manuscript.

  1. The writing must be improved, it is still very poor with numerous wrong terms, typos, or grammar mistakes (i.e. line 73, line 151, and so on).

Response:Thank you for your suggestion. According to your suggestion, we checked the manuscript and corrected the grammar mistakes in the revised manuscript.

  1. Lines 28-36: .....[4-8]........  [9-16].....[17-21]..... [23-27]......

You will likely need to re-write your citation sentences, rather than simply replace the numbers with Authors’ names.

Response:Thank you for your suggestion. According to your suggestion, we revised the introduction of the manuscript in the revised manuscript.

  1. Insert a picture with the “Static corrosion equipment” (line 88).

Response:Thank you for your suggestion. Fig 1 was the scheme of static corrosion equipment

  1. Line 94. Erase “equation-1” and write after the mathematical formula only equation number (1).

Response:Thank you for your suggestion. According to your suggestion, we deleted “equation” in the revised manuscript.

  1. Line 95: erase (unit:..). It is not necessary.

Response:Thank you for your suggestion. According to your suggestion, we deleted “unit” in the revised manuscript.

  1. Lines 98-103. Write complete information about tools (country, and so on).

Response:Thank you for your suggestion. According to your suggestion, we added information of characterization equipment in the revised manuscript.

  1. Give more details about the statistical analysis applied in this manuscript (method, software, validation, and so on).

Response:Thank you for your comments. When we carry out SEM characterization, we use the observation method of multi regions to observe the surface of the 316 SS samples after corrosion. the depth of corrosion holes and Cr depletion were the average of 5 position measurements. the result of weight change is the average of the weight changes of three parallel samples.

10, Lines 125-126: What does it mean “relatively smooth”. Can you specify some roughness parameters for this surface microtexture?

Response:Thank you for your suggestion. we deleted the sentence in the revised manuscript.

  1. Insert all the SEM parameters, such as magnification, acceleration voltage, working distance, and image pixel resolution (figs. 3 and 4).

Response:Thank you for your comments. When we use SEM to analyze the samples, the accelerating voltage selected by SEM is 10keV, and that selected by EDS is 20keV, the WD is 8-10 mm and the magnification is 1k times. Because the edges of the original images occupy a large blank space, most authors will treat the images as shown in Figures 3 and 4.

  1. Explain with more details sentences from pages: 148-149, 174-175, 252-255.

Response:Thank you for your suggestion. According to your suggestion, we added more detail sentence in line 148-149;174-175.252-255 in the revised manuscript.

  1. Lines 231-234. Erase “equation-2....5” and write after the mathematical formula only the equation number in parentheses.

Response:Thank you for your suggestion. According to your suggestion, we revised the lines 231-234 in the revised manuscript.

  1. Insert Author Contributions, Data Availability Statement, and Conflict of interest.

Response:Thank you for your suggestion. According to your suggestion, we added Author Contributions, Data Availability Statement, and Conflict of interest in the revised manuscript.

  1. Even though the work is relevant to the journal's scope, i.e., Materials, I do not find even a single article published in the journal in the list of references. Please, some references from MDPI journals.

Response: Thank you for your suggestion. According to your suggestion, we adjusted the cited references in the revised manuscript.

Round 2

Reviewer 1 Report

The revised manuscript has been enhanced; however, there are still points to improve, which are detailed in the following paragraphs:

1)    In the introduction, clarify which is “corrosion ability.” Also, the information should be complemented with information related to chloride molten salts at low temperatures.

2)    In the experimental section, include information about the preparation of samples analyzed by SEM and DRX. Clarify the origin of the impurities (why those elements/compounds and those concentrations?). Did you clean the samples just with water or another reagent?

3)    In the results and discussion, explain in detail the effect of temperature, and complement the discussion with other similar studies related to the mechanism of intergranular and breakaway. Why is it only showing the Element distribution at 600 °C? Please add the images at 500 and 700 °C. The DRX analysis should also be improved. Besides, labeling the valence states of ions in the reaction equation should be standardized.

4)    In the conclusion, complement the sentences to explain the effects of the temperature and impurities of molten salts.

Author Response

Thank you and all other reviewers for the critical feedback. We are truly grateful to yours and other reviewers’ critical comments and thoughtful suggestions. Based on these comments and suggestions, we have made careful modifications on the original manuscript. We hope the revised manuscript will meet materials standard. Following the reviewers and editor comments, we have modified the manuscript accordingly, and the detailed corrections are listed below point by point.

1.In the introduction, clarify which is “corrosion ability.” Also, the information should be complemented with information related to chloride molten salts at low temperatures.

Response: Thanks for the reviewer’s comment. the main salt constituents (KCl, MgCl2) are much more stable than potential corrosion products (CrCl2, NiCl2, FeCl2 etc.) from the thermal-dynamics calculation results, so we know that the corrosion of nickel or iron-based alloy in chloride salts such as pure molten KCl and MgCl2 is expected to be minimal. 

In order to avoid misunderstanding of the meaning of “corrosion ability”, I deleted it in the revised manuscript. At the same time, the corrosion rate at different temperatures reported in reference 21 is supplemented in the introduction section of the revised manuscript.

2.1   In the experimental section, include information about the preparation of samples analyzed by SEM and DRX.

Response: Thank you for your suggestion. According to your suggestion, we added the following information about the preparation of samples analyzed by SEM and XRD has been added in the revised manuscript.

“Scanning electron microscopy (SEM, Merlin Compact) coupled with an energy dispersive X-ray spectrometer (EDS) was used to analyze the surface and cross-sectional morphology changes of the corroded 316SS alloys. After cleaning and drying, the surface morphology and element analysis of the alloy samples could be carried out directly. Before the cross-section analysis, alloy samples need to be embedded with cold-curing epoxy resin, then ground to 1200 grit with silicon carbide paper and polished with 0.05 μm Al2O3 powder, after cleaning and drying, the cross-section morphology and element analysis of alloy samples could be carried out. The crystal phases of the cleaned and dried 316SS specimens before and after corrosion were measured by X-Ray diffraction (Bruker D8 Advance).

2.2 Clarify the origin of the impurities (why those elements/compounds and those concentrations?).

Response: Thanks for the reviewer’s comment. There are many kinds of impurities in KCl-MgCl2 salt. We only selectively determined some of them according to the impurity composition table of raw salts and the main elements of 316 SS alloy. Impurities of molten KCl-MgCl2 salts are derived from raw materials and/or from interactions during service process.

2.3 Did you clean the samples just with water or another reagent?

Response: Thanks for the reviewer’s comment. The corroded samples were cleaned with pure water and rinsed with anhydrous alcohol before drying. According to your suggestion, we added “rinsed with anhydrous alcohol” in the revised manuscript.

  “Clean the corroded 316SS alloys with deionized water, rinsed with anhydrous alcohol and dried it with cold air”

3.1 In the results and discussion, explain in detail the effect of temperature, and complement the discussion with other similar studies related to the mechanism of intergranular and breakaway.

Response: Thanks for the reviewer’s comment. Based on reviewer’s suggestion, we revised the discussion. The following was the revisions in the revised manuscript.

“In chemical reaction, the reaction rate is directly related to the concentration of reactants, and the higher the concentration of oxidizing impurities in salt, the faster the dissolution rate of Cr and Fe. Mg reducing agent was used to purify KCl-MgCl2 eutectic salt, which reduced the concentration of oxidation impurities, thus reducing the corrosion of 316 SS. When Cr and Fe in the outermost layer of the 316SS alloy are dissolved, the Cr and Fe concentrations on the surface of the 316SS are different from those in the matrix. Driven by the concentration gradient, Cr and Fe in the matrix diffuse from matrix to the surface of 316SS, and continued to react with impurities in KCl-MgCl2 eutectic salt, resulting in continuous corrosion of the alloy.

A large number of studies showed that corrosion rate of Cr and Fe is not only related to the concentration gradient, but also related to the diffusion rate of Cr and Fe atoms [6,24]. The results show that the grain boundary diffusion rate of Cr in 316SS is about 106 times of lattice diffusion rate, which indicates that grain boundary diffusion is much faster than lattice diffusion [25]. Therefore, after 100 hours of corrosion test, all the corroded 316SS samples showed different degrees of grain boundary corrosion.

The diffusion of Cr and Fe in 316SS can be described by[26]:

D = D0e(-Q/RT)      (2)

where D is the diffusion coefficient, D0 is the material coefficient, Q is the diffusion activation energy, R is the inert gas constant, and T is the Kelvin temperature. With the increase of temperature, the diffusion coefficient of atoms increases, and the diffusion rate of Fe and Cr atoms from 316 SS matrix to surface increases, which leads to resulting in the increase of intergranular corrosion rate of 316SS, which is consistent with the corrosion trend of 316SS in NaCl-MgCl2 salt [21].

It is found that impurity content in salt and salt temperature will affect alloy corrosion [17-22]. The influence of impurities on the corrosion rate of the 316SS alloy was not as significant as that of temperature, which was due to the impurity content in the unpurified KCl-MgCl2 eutectic salt was not much higher than that in the purified salt, and the corrosion tests were carried out in an inert atmosphere. The different of impurities concentration between salts was limited. This might also be the main reason why the effect of purification on 316SS corrosion is less than that of temperature. Meanwhile, the corrosion process of alloy in purified salt is largely determined by the diffusion rate of corrosive atoms such as chromium in the matrix to the outermost layer of alloy, which increases linearly with the increase of temperature [25], so the temperature has a great influence on the corrosion of 316SS.”

3.2 Why is it only showing the Element distribution at 600 °C? Please add the images at 500 and 700 °C.

Response: Thanks for the reviewer’s comment. Based on reviewer’s suggestion, we revised the Fig.5 and added the mapping results of 316SS (B2) exposed in purified salt under Ar for 100 hrs at 500°C (B1) and 700°C (B3).

3.3The DRX analysis should also be improved.

Response: Thanks for the reviewer’s comment. Based on reviewer’s suggestion, we revised the Fig.7(XRD). At the same time, we modified the analysis of the XRD results in section 3.4. The following was the revisions in the revised version.

Figure 7. XRD patterns of the original 316SS (a) and corroded 316SS in salt A (A1,A2,A3) and salt B (B1,B2,B3) under Ar for 100h at 500°C(A1,B1),600°C (A2,B2) and 700°C (A3,B3).

“XRD patterns of the original and corroded 316SS in the unpurified and purified KCl-MgCl2 salt at different temperatures were shown in Fig.7. Before corrosion, the characteristic diffraction peaks of 316SS, as shown in black XRD pattern, indicates that the initial microstructure of 316SS alloy is single phase face centered cubic (FCC) austenite structure. After corrosion, the crystal structure of the corroded 316SS remained austenite structure. Only the diffraction peaks of the 316SS corroded in salt slightly moved to a higher angle, which is attributed to the dissolution of Fe and Cr on the surface of 316SS [14,21]. Except for the matrix peak of 316 SS, a new characteristic peak appeared on the surface of the alloy after corrosion. Combined with the analysis of SEM, it could be inferred that it was the characteristic peak of MgO. This shows that MgO appeared on the surface of the corroded alloy.

The results of XRD show that the basic crystal structure of 316SS alloy surface will not change with the change of temperature and purity after corrosion, However, when the salt temperature and salt purity change, the shift of characteristic peak position of 316SS alloy matrix to high angle is slightly different.”

3.4 labeling the valence states of ions in the reaction equation should be standardized.

Response: Thanks for the reviewer’s comment. Based on reviewer’s suggestion, we labeling the valence states of ions in the reaction equations.

4) In the conclusion, complement the sentences to explain the effects of the temperature and impurities of molten salts.

Response: Thanks for the reviewer’s comment. Based on reviewer’s suggestion, we modified the conclusion section and added the following sentences (blue color) to explain the effects of the temperature and impurities of molten salts.

“The effect of temperature and purity on the corrosion of 316SS in KCl-MgCl2 salt were studied through static immersion corrosion under Ar at 500°C, 600°C and 700°C for about 100 hrs. The results showed that the corrosion rate of 316SS in KCl-MgCl2 salt increased nonlinearly with the increase of temperature. Below 600 ℃, the corrosion rate of 316SS increased slowly with the increase of temperature. When the salt temperature rose to 700℃, the corrosion rate of 316SS increased dramatically. Purification can reduce the corrosivity of KCl-MgCl2 salt. At these three temperature points, the corrosion of 316SS by KCl-MgCl2 salt after Mg metal reduction and purification weakened.”

Sincerely yours,

Zhongfeng Tang. Ph.D, Prof.

Shanghai Institute of Applied Physics, Chinese Academy of Sciences, Jialuo Road 2019, Jiading, Shanghai 201800, China

Tel:+ 86-21-39194681
